# Modulated Electro-Hyperthermia Resolves Radioresistance of Panc1 Pancreas Adenocarcinoma and Promotes DNA Damage and Apoptosis In Vitro

**DOI:** 10.3390/ijms21145100

**Published:** 2020-07-19

**Authors:** Gertrud Forika, Andrea Balogh, Tamas Vancsik, Attila Zalatnai, Gabor Petovari, Zoltan Benyo, Tibor Krenacs

**Affiliations:** 11st Department of Pathology and Experimental Cancer Research, Semmelweis University, 1085 Budapest, Hungary; forika.gertrud@med.semmelweis-univ.hu (G.F.); zalatnai.attila@med.semmelweis-univ.hu (A.Z.); petovari.gabor@med.semmelweis-univ.hu (G.P.); 2Institute of Translational Medicine, Semmelweis University, 1094 Budapest, Hungary; balogh.andrea@med.semmelweis-univ.hu (A.B.); vancsik.tamas@med.semmelweis-univ.hu (T.V.); benyo.zoltan@med.semmelweis-univ.hu (Z.B.)

**Keywords:** modulated electro-hyperthermia (mEHT), Panc1 pancreas adenocarcinoma, caspase-dependent apoptosis, p21^waf1^ upregulation, resolution of radioresistance

## Abstract

The poor outcome of pancreas ductal adenocarcinomas (PDAC) is frequently linked to therapy resistance. Modulated electro-hyperthermia (mEHT) generated by 13.56 MHz capacitive radiofrequency can induce direct tumor damage and promote chemo- and radiotherapy. Here, we tested the effect of mEHT either alone or in combination with radiotherapy using an in vivo model of Panc1, a KRAS and TP53 mutant, radioresistant PDAC cell line. A single mEHT shot of 60 min induced ~50% loss of viable cells and morphological signs of apoptosis including chromatin condensation, nuclear shrinkage and apoptotic bodies. Most mEHT treatment related effects exceeded those of radiotherapy, and these were further amplified after combining the two modalities. Treatment related apoptosis was confirmed by a significantly elevated number of annexin V single-positive and cleaved/activated caspase-3 positive tumor cells, as well as sub-G1-phase tumor cell fractions. mEHT and mEHT+radioterapy caused the moderate accumulation of γH2AX positive nuclear foci, indicating DNA double-strand breaks and upregulation of the cyclin dependent kinase inhibitor p21^waf1^ besides the downregulation of Akt signaling. A clonogenic assay revealed that both mono- and combined treatments affected the tumor progenitor/stem cell populations too. In conclusion, mEHT treatment can contribute to tumor growth inhibition and apoptosis induction and resolve radioresistance of Panc1 PDAC cells.

## 1. Introduction

The main cause of mortality and morbidity of pancreatic cancer is due to pancreatic ductal adenocarcinoma (PDAC). PDAC, which represents ~85% of the pancreatic cancers [1], has been associated with very poor prognosis of an estimated <10% five-year survival rate [2]. This poor outcome rate reflects the ineffectiveness of the current therapies, especially when the tumor is unresectable. The best treatment results of PDAC have been achieved by surgical resection preceded by neoadjuvant therapy [3] but only 10%–20% of diagnosed pancreatic cancers are resectable [4]. Disposable neoadjuvant and adjuvant therapies are represented by chemotherapy (Gemcitabine, FOLFIRINOX or nab-Paclitaxel), radiotherapy or their combinations [3]. Hyperthermia has been suggested to improve the efficiency of both radio- [5] and chemotherapy but its mechanism in PDAC treatment requires further clarification, which we tested in this study using modulated electro-hyperthermia (mEHT) treatment.

Some studies have demonstrated improved survival chances in patient groups receiving chemoradiotherapy versus those treated with chemotherapy only [6,7,8], while others have shown no benefit or even a higher toxicity when chemotherapy was supplemented with radiotherapy [9,10]. These contradictory data may generally be attributed to the radioresistance of PDAC tumors. Despite their morphological similarities, tumors even within a diagnostic group may display vast inter- and intratumoral heterogeneity [11,12]. The bulk of tumor cells usually represents the proliferating “transit amplifying” population, which is sensitive to cytoreductive therapies [13]. However, the presumed tumor stem-/progenitor cells, though constantly self-review, rarely replicate, which makes them hidden targets for chemo- and radiation therapies which leads to treatment resistance [14,15]. This tumor heterogeneity and the related treatment restraints have also been confirmed in PDAC [16].

The limited efficiency of traditional therapies forced the development and clinical application of complementary techniques which can improve the success of the accessible anticancer treatments. High fever-range (41–42 °C) hyperthermia has been described as one of the most potent radiosensitizers [17]. Numerous clinical studies have shown its benefits in local tumor control and survival outcome when combined with irradiation compared to single radiotherapy [18,19,20,21]. The poor vascularization of malignant tumors can lead to oxygen and nutrient deprivation and reduced pH levels, which may hinder the success of irradiation [22]. Hyperthermia is thought to improve tumor perfusion and oxygenation to counterbalance these factors [23]. Furthermore, in some tumors, hyperthermia can block DNA repair enzymes to further support the cytotoxic effect of radiation induced DNA breaks [24,25,26,27].

In PDAC, an increasing number of clinical studies have demonstrated that different forms of high-fever range hyperthermia including WBH (whole body hyperthermia), HIPEC (hyperthermic intraperitoneal chemotherapy) and locoregional delivery may improve the effects of chemotherapy both concerning survival and quality of life parameters [28,29,30]. Recent evidences have also shown the clinical benefits of mEHT (Oncothermia) in PDAC treatment in combination with chemotherapy [31,32,33]. These data led to the initiation of clinical trials combining PDAC chemotherapy with locoregional hyperthermia generated by instruments using different radiofrequency and power ranges based either on radiative (70–120 MHz, 800–1500 W, 60 min/treatment) [34] or capacitive (mEHT: 13.56 MHz, 60–150 W, 60 min/treatment) [35] delivery. A better understanding of the mechanism of hyperthermia in PDAC treatment may help to design more efficient treatment protocols.

Modulated electro-hyperthermia delivers loco-regional deep hyperthermia by using 13.56 MHz radiofrequency. The generated electric field can be concentrated in malignant tissue to induce a selective temperature increase (controlled at 42 °C), as a result of elevated glycolysis, lactate concentration and electric conductivity compared to the adjacent tissues [36]. We have confirmed earlier in preclinical studies, that mEHT even in monotherapy can provoke tumor damage, which can be further enhanced in combination with immune promoting or chemotherapeutic agents [37,38]. In monotherapy, besides inducing DNA damage and caspase-dependent apoptosis, mEHT also inhibited tumor cell growth and proliferation in colorectal cancer both in vitro and in vivo and promoted the uptake and cytotoxicity of the anthracycline antibiotic Doxorubicin, in vitro [37].

In this study, we set up an in vitro model with a Panc1 pancreatic adenocarcinoma cell line and tested the efficiency and response of tumor cells to mEHT alone and in combination with radiotherapy.

## 2. Results

### 2.1. Tumor Cell Loss Induced by mEHT Alone and in Combination with Irradiation

Treatments for 30, 60, 120 and 240 min, respectively, using mEHT at 42 °C were tested to define an LD50 in Panc1 monolayers. Based on the hematoxylin and eosin (H and E) stained tumor cell cultures, 60 min mEHT treatment caused ~50% cell loss and morphologically visible changes suggestive primarily of programmed cell death response in ~20% of the remaining adherent cells (Figure 1A,B). These signs included nuclear shrinkage, chromatin condensation and the formation of apoptotic bodies. 

As opposed to this, 2 Gy irradiation resulted in only moderate cell loss compared to the untreated control groups. The number of viable tumor cells counted in the detached cell fractions after trypan blue staining was significantly reduced both after mEHT (*p* < 0.05) and R+mEHT (*p* < 0.01) treatments compared to the control groups (Figure 1C). The therapeutic enhancement ratio (TER)—defined as the ratio of radiation sensitivity at an elevated temperature compared to that achieved at 37 °C [39] showed a TER of = 1.86 after mEHT.

### 2.2. Apoptosis Dominated Cell Death

The analysis of annexin V and propidium iodide (PI) double staining flow cytometry allowed the identification and distinction of the live, apoptotic and necrotic cell populations (Figure 2A). Apoptotic cells and bodies took up only annexin V but no PI, while necrotic cells were either double-positive or stained only with PI. Accordingly, consistent with the morphological signs, the damage caused by mEHT monotherapy was apoptosis dominated programmed cell death. The Kruskal–Wallis test revealed a significant difference (*p* = 0.0084) among treatment groups in the amount of the early apoptotic bodies, however, the Dunn’s post hoc test did not reach significance in pairwise comparisons concerning the mEHT and R+mEHT groups compared to the controls. The TER instead had a value of 0.28 which highlights the additive effect of hyperthermia to radiation therapy. This is in line with the cell viability test, while irradiation monotherapy showed no difference compared to the controls. (Figure 2B).

Immunohistochemistry revealed a major elevation in the number of cleaved caspase-3 positive tumor cells after treatment regimens involving mEHT, which was confirmed by image segmentation based digital image analysis (Figure 2C). Significantly more adherent tumor cells with nuclear cleaved/activated caspase-3 positivity were counted after R+mEHT treatment than either in the untreated controls or radiotherapy alone. The mean proportions of the positive cells showed a progressive increase from 2.88 ± 1.18 in the control groups, through 8.28 ± 2.22 after mEHT monotherapy, to 17.01 ± 1.2 after combined R+mEHT treatment. The effect of irradiation alone remained at the level of the untreated controls, supporting the radioresistance of the Panc1 cell line (Figure 2D). Capillary Western blot analysis showed a discrete increase in caspase 8 and caspase 9 protein levels in the mEHT and R+mEHT treated groups compared to the control, but the elevation was not statistically significant. The STAT3 and BAX protein levels did not show any changes between the 4 groups.

### 2.3. Treatment Related Reduction in Tumor Progenitor/Stem Cell Colonies

Colony forming ability and dynamics are thought to be reliable markers of cell renewal capacity. Single cells that grow into clones of at least 50 cells in 1–3 weeks after treatment represent tumor progenitor/stem cells [40]. We tested the colony formation ability of Panc1 cells by seeding 10^4^/10 mL pretreated cells into 100 mm Petri dishes and counted the crystal violet stained colonies after 8 days of growing in culture media (Figure 3A,B). Combined R+mEHT treatment significantly reduced the number of tumor colonies compared to the untreated controls (*p* = 0.019), while the statistical analysis among the 4 groups showed a strong difference (*p* = 0.0064) (Figure 3C).

### 2.4. Treatment Related Inhibition of p-Akt/Ser473 and p21^waf1^ Upregulation

Phosphorylation of Akt at Ser473 correlated with cell growth, proliferation and survival. A strong tendency of reduced p-Akt/Ser473 levels was detected after mEHT monotherapy and the combined R+mEHT treatment. A statistically significant difference was detected among the 4 groups using the Kruskal–Wallis test (*p* = 0.015) (Figure 4A,B). Since Akt activated by phosphorylation can inhibit p21^waf1^ tumor suppressor protein, its levels were also tested. As a potential link with reduced p-Akt/Ser473 levels, Western blot analysis showed significantly increased p21^waf1^ protein expression in tumor cell cultures treated with R+mEHT compared to the controls (*p* = 0.013) and the Kruskal–Wallis test showed a strong significance among the 4 groups (*p* = 0.0014) (Figure 4C,D).

### 2.5. Treatment Related Increased of SubG1-Phase Tumor Factions

Presuming that the damaged DNA can delay and/or block cell replication, the cell cycle phases were also evaluated after treatments. The significantly elevated SubG1-phase fractions, after combined R+mEHT treatment, were in line with treatment related apoptosis (Figure 5A,B). This was accompanied by reduced G1-phase cell fractions, but without a major difference either in the S- or G2-phase cell populations within the replication cycle. The number of tumor cells with elevated nuclear foci of the phosphorylated H2AX protein (Figure 5C), indicating DNA double-strand breaks increased after all treatment regiments, however, these values did not reach up to significance (Figure 5D,E).

## 3. Discussion

Radio- and chemoresistance are major obstacles imposed on the successful clinical management of advanced PDAC cases [41,42]. Preclinical studies have demonstrated that mEHT even in monotherapy can induce irreversible cell stress leading to apoptosis and reduced tumor masses in different tumor models [38,43,44,45]. In mouse colorectal adenocarcinoma we have shown that mEHT, when combined with a T-cell promoting agent, can support the antitumor immune response even at a distant location from the treated site (abscopal effect), and it can promote the uptake and efficiency of doxorubicin chemotherapy [37,38]. Though traditional hyperthermia has been shown to reduce the motility and thus the invasion of Panc1 pancreas adenocarcinoma cells [46,47] or even cause cell death, the latter was tested only under non-conventional treatment conditions. In mEHT, in addition to the generated heat, the electric field can interfere with charged molecules including death-signaling related ones, in tumor cell membranes and can contribute to additional “non-thermal antitumor effects” [48]. There has been very limited information available on how mEHT can contribute to tumor damage in pancreas adenocarcinoma. Since a better understanding of the mechanism of action can support the design of improved treatment protocols, here, we tested if mEHT induced hyperthermia can add to the efficiency of irradiation in a PDAC model using the TP53 mutant and known radioresistant Panc1 tumor cells. 

By testing different durations of mEHT monotherapy, 60 min treatment led to a ~50% tumor cell loss (LD50) which, therefore, was selected for further studies. This mEHT treatment of Panc1 pancreas adenocarcinoma cells resulted in a very similar anti-tumor effect that we achieved earlier after 2 × 30 min treatment in mouse colon adenocarcinoma (C26) cell cultures [37]. These involved an irreversible cell stress and apoptosis, which were also confirmed in a C26 allograft model [38]. In Panc1 cell cultures, the mEHT-related cell loss was accompanied by a massive reduction in viable tumor cells, which was further pronounced when mEHT was combined with radiotherapy. The higher number of the remaining adherent tumor cells showed a sign of apoptosis related programmed cell death in these groups compared to 2Gy irradiation monotherapy. These results reflect the well documented high radioresistance of Panc1 cells [49,50] and the ability of mEHT to resolve this resistance when combined with radiotherapy. 

In the Panc1 cell line, high-fever range hyperthermia has been described to reduce (TGF-β1-induced) tumor migration, and epithelial-mesenchymal transition (EMT) driven invasion both in genetically unmanipulated—using 43 °C for 60 min, [47] and in gemcitabine resistant Panc1 pancreas ductal adenocarcinoma cells—at 42 °C for 60 min [46]. However, limited information was available on the destructive effect of hyperthermia on Panc1 tumor cells. One group showed that both extrinsic (hot air and water bath), or iron oxide nanoparticle based magnetic hyperthermia can damage collagen architecture in hetero-spheroids of Panc1 cells and WI-38 fibroblasts leading to cell death, but without specifying the contribution of Panc1 to this [51]. Another study demonstrated mitochondrial-driven apoptosis of Panc1 tumor cells after using PCR thermal cycling-based heating to 44 °C, which, however, was well above the conventional 42 °C therapeutic temperature [52]. In our present study, mEHT monotherapy at 42 °C and mEHT in combination with radiotherapy provoked a significantly elevated apoptosis. Besides chromatin condensation, nuclear shrinkage and apoptotic bodies, the programmed cell death response was also confirmed by the elevated subG1 cell fraction after treatments. Additionally, the annexin V-PI flow cytometry analysis showed the prevailing of apoptotic tumor cell fraction over the necrotic except after irradiation monotherapy, suggesting different killing mechanisms by these treatments. The elevated number of tumor cells with cleaved/activated caspase-3 positive nuclei was consistent with a dominant caspase-dependent apoptosis. As in most mEHT-treated tumor models, both the extrinsic and intrinsic pathways were likely to be involved here.

As a KRAS mutated cell line, survival mechanisms are upregulated per se in Panc1 cells involving elevated Akt expression and activation [53]. In line with earlier studies by other groups, we also confirmed relatively high activated p-Akt/Ser473 levels in Panc1 cells using flow cytometry [54,55]. This showed a trend of downregulation both after mEHT monotherapy, and when mEHT was combined with radiotherapy. A similar effect of mEHT treatment on p-Akt/Ser473 levels was detected by our group in a colorectal adenocarcinoma model in combination with doxorubicin [37]. In Panc1 cells, the treatment-related p-Akt/Ser473 reduction was accompanied by a significant elevation in p21^waf1^ protein expression. The upregulation of p21^waf1^ has also been recognized both in colorectal cancer (C26) and melanoma (B16F10) treated with mEHT, suggesting that this cycling dependent kinase inhibitor may play an important role in tumor mass reduction by mEHT [37,43]. However, we could not detect significantly increased G1 or G2/M phase cell fractions expected from a p21^waf1^ mediated delayed cell cycle progression [56]. The possible reason for this may lie in the massive elevation in the apoptotic, subG1-phase cell fractions reflecting a pronounced direct tumor damage by mEHT. 

The treatment-related tumor damage was not concentrated only on the bulk of proliferating tumor cells, known as the transit amplifying cell population, which can usually be interfered by traditional chemo- and/or radiotherapy [14,15], but also targeted and reduced the tumor progenitor/stem cell populations. A colony forming assay in low density Panc1 cells grown for 8 days showed less colonies after mEHT than post-irradiation, whose decline became dramatically more efficient up to the level of synergism when these two treatments were combined. This again underlined the ability of mEHT treatment to support radiotherapy but without the known mechanism of hyperthermia through improved reoxygenation, which is not likely to play a major role in this in vitro model.

## 4. Materials and Methods

### 4.1. Cell Culturing and Irradiation

The Panc1 pancreas adenocarcinoma cell line (ATCC, Teddington, Middlesex, UK) was grown in Dulbecco’s Modified Eagle Medium (DMEM LM-D1111/500, Biosera, Boussens, France) enriched with 10% heat inactivated fetal bovine serum (FBS FB-1090/500, Biosera, Boussens, France). For treatments, the cells were harvested using 0.25% trypsin and 0.22 mg/mL ethylenediaminetetraacetic acid (Trypsin-EDTA, XC-T1717/100, Biosera, Boussens, France) and centrifuged at 300× *g*. Then, 4 × 10^5^ cells were resuspended in 4 mL fresh culture medium and seeded into 60 mm Petri dishes containing a sterile 24 × 40 mm coverslip. Cells were kept in a humidified incubator at 37 °C with an atmosphere containing 5% CO_2_.

Panc1 cells cultured in Petri dishes were irradiated with GSR D1 gamma irradiator, using Cs-137 isotope (Gamma-Service Medical GmbH, Leipzig, Germany) and 2 Gy irradiation dose per sample.

### 4.2. mEHT Treatment

The Panc1 cultures grown on coverslips were immersed into a 10 mm wide vertical glass container, filled with 60 mL cell culture medium of 35–37 °C, which was placed between the two electrodes of the Lab-EHY 100 device (Oncotherm Kft, Budaors, Hungary). First, a pre-heating was applied for 5 min using 20–25 W power to reach 42 °C, then the device was adjusted between 7 and 8 W to maintain this temperature for 60 min, with the internal temperature continuously monitored by using two glass-fiber sensors immersed into the medium. After treatment, the coverslip cultures were kept in fresh medium and grown as before under normal conditions (see above) for 24 h.

### 4.3. Cell Viability Test

According to the principle, the trypan blue dye cannot pass through intact cell membranes; thus, viable and damaged cells can be easily distinguished under a light microscope. A total of 24 h after treatment, the cells were trypsinized and gently agitated to obtain one-cell suspensions. A small amount (20 µL) of the suspension was mixed with 5 µL Trypan blue solution (0.4% Sigma-Aldrich, Inc., St. Louis, MO, USA) and transferred into a Bürker chamber. Viable cells were counted under a light microscope.

### 4.4. Cytomorphology and Immunocytochemistry

Treatment related morphological changes were studied after hematoxylin and eosin (H and E) staining. Coverslip cultures were fixed in 10% formalin for 15 min and stained with hematoxylin for 30 s followed by gentle washing and blueing in tap water, then stained in eosin for 3 min. 

For immunocytochemistry, the formalin-fixed coverslip cultures were washed in TBST puffer made from 0.01 mol/L tris-buffered saline pH 7.4 (TBS) and 0.1% Tween-20 (Fisher Scientific UK Ltd., Loughborough, UK). Then, permeabilization was performed for 30 min in TBS containing 0.3% Tween-20. Before immunoreactions, endogenous peroxidases were quenched in methanol containing 3% hydrogen peroxide. Nonspecific protein binding sites were then blocked for 20 min using TBST buffer containing 3% bovine serum albumin (Probumin, BSA, 82-100-6, Merk, Darmstadt, Germany). Rabbit monoclonal antibodies diluted in TBST were used overnight (16 h) to detect cleaved caspase-3 (1:300, clone: 5A1E, #9664) and phospho(Ser139)-histone γ-H2AX (1:150, clone: 20E3, #9718) both from Cell Signaling (Danvers, MA, USA) followed by using polymer-peroxidase labeled mouse and rabbit IgGs (Histols MR-T, Histopathology Ltd., Pecs, Hungary), for 40 min. All steps were performed at room temperature and the coverslip cultures were washed in TBST for 2 × 3 min between the incubation steps. The chromogen reaction was revealed using a DAB chromogen/hydrogen peroxide kit (DAB Quanto, TA-060QHDX, Thermo-Fisher, Cheshire, UK). Cell nuclei were counterstained with hematoxylin. All stained coverslip cultures were dehydrated in graded ethanol series, mounted onto glass slides from xylene, digitalized (Pannoramic scanner, 3DHISTECH, Budapest, Hungary) and analyzed using the QuantCenter image analysis (Figure 6) software package (from 3DHISTECH, Budapest, Hungary).

### 4.5. Flow Cytometry

Cells released from the coverslips by trypsinization were neutralized and harvested in their supernatants, washed in phosphate-buffered saline (PBS) and collected in FACS tubes. After centrifugation at 300× *g* for 8 min, the number of cells resuspended in PBS was determined in a Bürker chamber with a minimum of 10^5^ cells to be used for each staining. For apoptosis testing, immediate staining with Alexa Fluor 647 annexin V (1:20; #640912, BioLegend, San Diego, CA, USA) and 1 µL of 1 mg/mL stock solution (diluted in PBS) of propidium iodide (PI, 1304MP, Thermo-Fisher, Cheshire, UK) for 15 min was performed at room temperature in the dark. Measurements were performed after adding 300 µL annexin-V-binding buffer (#422201 BioLegend, San Diego, CA, USA).

For testing activated/phosphorylated Akt levels, free-floating cells were washed in PBS, centrifuged at 300× *g* for 8 min and fixed in 10% neutral buffered formalin at 4 °C for 30 min. After washing twice in PBS for 20 min, permeabilization in 0.2% Tween/PBS and rigorous washing and centrifugation were performed again. The cells were incubated with a monoclonal rabbit anti-phospho-Akt(p-Akt/Ser473) antibody (1:100, clone: D9E; Cell Signaling, Danvers, MA, USA) for 60 min, washed and incubated again for 30 min with Alexa Fluor 488 conjugated anti-rabbit Igs (1:100; Thermo Fisher, Cheshire, UK), then washed, centrifuged and collected in 300 µL PBS for analysis.

For cell cycle analysis, tumor cell suspensions were fixed in ice-cold ethanol for 60 min, washed and span down twice in PBS, and then incubated with 250 µL PBS containing 20 ng RNaseA (R5503, Sigma-Aldrich, Inc., St. Louis, MO, USA) and 10 µL of 1 mg/mL PI stock solution at 4 °C for 60 min. 

Flow cytometry was performed on a CytoFLEX Flow Cytometer using its CytExpert software (Beckman Coulter, Indianapolis, IN, USA).

### 4.6. Clonogenic Assay

Low density 10^4^ tumor cells, released by trypsinization, were plated into 100 mm diameter Petri dishes 24 h after any treatment and grown under standard culture conditions. After 8 days, when colonies were clearly visible, the cell culture medium was removed, cells were washed in PBS and dried at room temperature. Tumor cell colonies were revealed using 0.1% crystal violet solution for 30 min. Manually counted cell groups containing at least 50 cells were considered to be colonies. 

### 4.7. Capillary Western Blot

Proteins were extracted using an extraction buffer made freshly by mixing 20 mM Tris, 2 mM EDTA, 150 mM NaCl, 1% Triton X-100; supplemented with 10 mM NaF, 0.5 mM NaVO3 and 1:200 Protease Inhibitor Cocktail (P8340, Sigma-Aldrich, St. Louis, MO, USA), as previously described [57]. Briefly, cells were washed with PBS, a 150 µL extraction buffer was added and the lysate was collected with a cell scraper. The mixture was incubated for 30 min on ice, then centrifuged at 4 °C and 12,000 rpm for 20 min. The total protein was quantified by the Bradford reagent (#500-0205, BioRad, Hercules, CA, USA) and the samples were kept at −20 °C until WES Simple analysis. 

The WES capillary Western blot device (ProteinSimple, San Jose, CA, USA) was used with the 12–230 kDa Jess/Wes separation module kit (SM-W004) according to the manufacturer’s instructions. The kit included a 25-capillary cartridge (12–230 kDa), pre-filled microplates with split running buffer, wash buffer, 10× sample buffer and an EZ standard pack with a 12–230 biotinylated ladder, a fluorescent 5× master mix and a dithiothreitol (DTT) containing tube. The lyophilized DTT and biotinylated ladder were suspended in the right amount of distilled water, the fluorescent 5× master mix was suspended in a solution of 20 µL previously prepared DTT solution and 20 µL 10× Sample Buffer. The samples were diluted to a concentration of 0.5µg/µl in 100× diluted ”10× Sample Buffer”, then 5 parts of sample solution were mixed with 1 part of fluorescent 5× master mix, followed by heating for 5 min at 95 °C. Mouse monoclonal anti-p21^waf1^ (1:25 clone CP47, NeoMarkers, Fremont, CA, USA), bcl-2 (1:50, #15071, Cell Signalling) and bax (1:50, #MS-711-P0, NeoMarkers) antibodies; rabbit monoclonal β-actin (1:50 13E5, Cell Signalling), STAT3 (1:50, #RB-9237-P0, NeoMarkers) and caspase 9 (1:50, #9502, Cell Signalling) antibodies as well as a rabbit polyclonal caspase 8 antibody (1:50, #JM-3158-100, MBJ, Woburn, MA, USA) were used. Peroxidase conjugated anti-rabbit and anti-mouse detection module reagents (ProteinSimple DM-001 and DM-002) followed by the manufacturer’s chemiluminescent substrate. The plates with components pipetted were centrifuged at 1000 g for 5 min, the fully prepared cartridges were inserted into the instrument which was used with the default settings of the software: stacking and separation at 395 V for 30 min; blocking reagent for 5 min, primary and secondary antibodies both for 30 min; luminol/peroxide chemiluminescence detection for 15 min (exposure times were between 1 and 512 s). The electropherograms were checked then the automatic peak detection was manually corrected if it was required.

### 4.8. Statistics

All results analyzed represented the means of at least 3 independent experiments. For the statistical analysis, the GraphPad Prism software package (San Diego, CA, USA) with the integrated tests was used: the non-parametric Kruskal–Wallis test and the Dunn’s multiple comparison post-hoc test were used. Significance was considered at *p* < 0.05 at a CI = 95%. All experiments were executed at least in triplicates.

## 5. Conclusions

In conclusion, our results show that mEHT can induce direct tumor damage and contribute to the effective radiotherapy of the resistant Panc1 pancreatic adenocarcinoma through DNA double-strand beaks and caspase dependent apoptosis. Additional tumor growth and survival inhibition by mEHT was mediated through the upregulation of p21^waf1^ and downregulation of Akt signaling. These affected not only the proliferating tumor bulk but also the tumor progenitor/stem cell population and significantly improved the efficiency of radiotherapy when combined with mEHT. All these data suggest that mEHT may help to resolve the radioresistance of this highly deregulated, TP53 and KRAS mutant PDAC, which needs to be further investigated using in vivo tumor models.

## Figures and Tables

**Figure 1 ijms-21-05100-f001:**
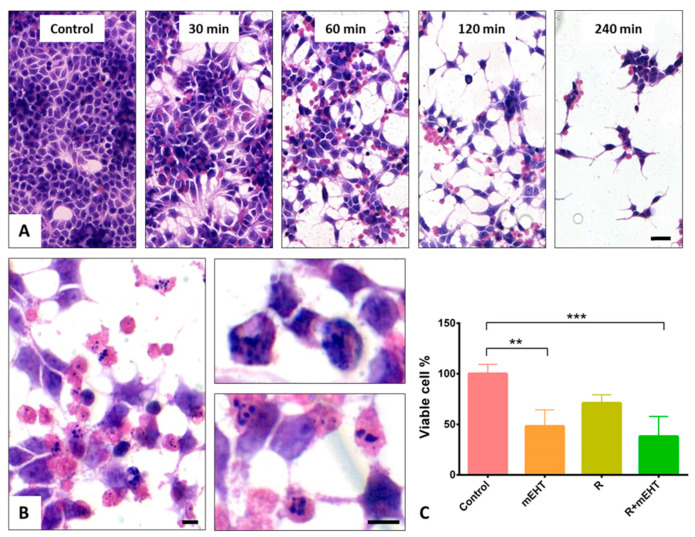
Hematoxylin and eosin staining of Panc1 cell cultures demonstrating cell loss and the major changes in cell morphology 24 h after different mEHT treatment durations (**A**). Scale bar: 40 µm. Higher magnifications reveal nuclear shrinkage, chromatin condensation (upper right) and apoptotic bodies (lower right) after 60 min treatment (**B**). Scale bars: 10 µm. Significant reduction in viable cell numbers, detected using trypan blue staining, both after 60 min mEHT monotherapy (*p* = 0.014), and after combination of mEHT and 2 Gy radiation (R+mEHT) (*p* = 0.0029) but not after radiotherapy alone (R). Kruskal–Wallis test reveals significant difference (*p* = 0.0007) among the 4 groups (**C**). The asterisks mark the value of *p*: ** means *p* ≤ 0.01, *** means *p* ≤ 0.001.

**Figure 2 ijms-21-05100-f002:**
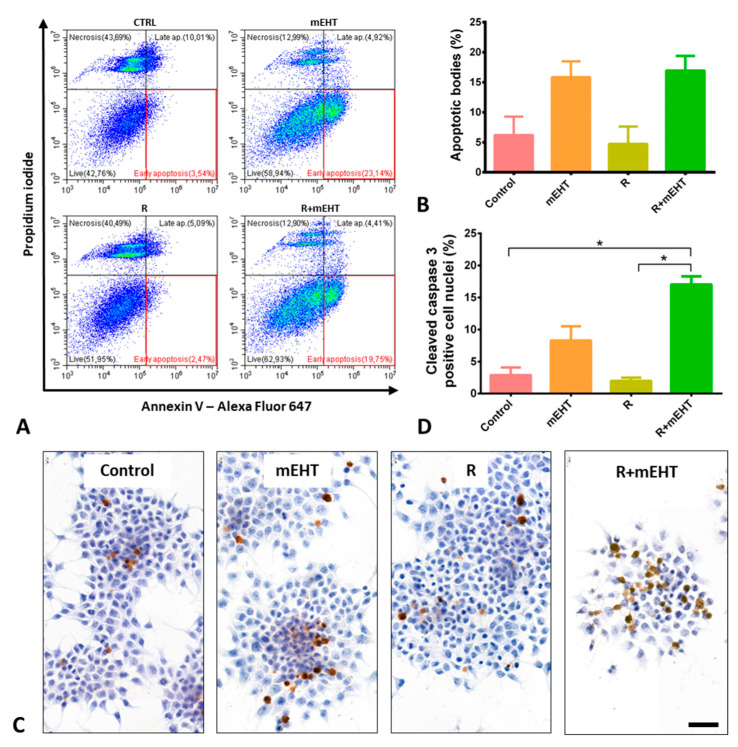
Apoptosis dominated programmed cell death induced by 60 min mEHT in Panc1 cultures. Representative flow cytometry results of Annexin V and propidium iodide double stained tumor cells 24 h after treatments. Red frames highlight early apoptotic cells (**A**). Graphical representation of early apoptotic bodies in proportions of the whole population. The Kruskal–Wallis test reveals a significant difference among the 4 groups: *p* = 0.0084 TER = 0.28 (**B**). Cleaved/activated caspase-3 immunoreactions (**C**) quantified as detailed in Figure 6 (scale bar = 50 µm) and graphical representation of its results. Kruskal–Wallis test shows a strong significance between the 4 groups: *p* = 0.0002. The Dunn’s test showed significantly elevated positivity in the R+mEHT group compared to R (*p* = 0.028) and compared to the control as well (*p* = 0.047) TER = 0.11 (**D**). The asterisk (*) marks the value of *p* ≤ 0.05.

**Figure 3 ijms-21-05100-f003:**
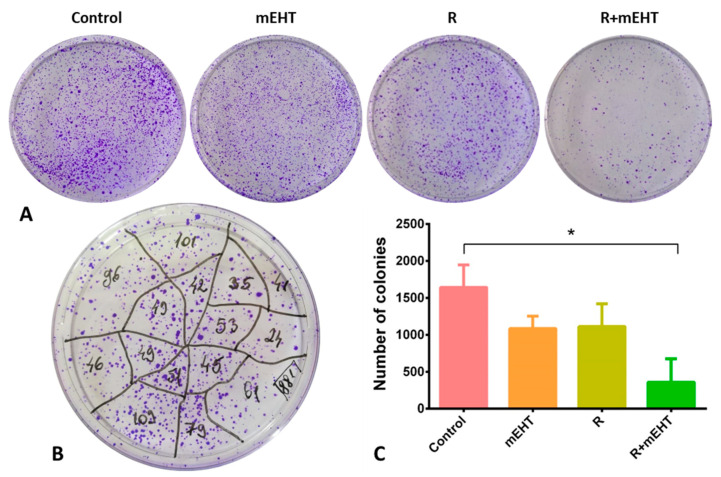
Clonogenic assay after different treatments of Panc1 tumor cells. Representative images of formalin-fixed, crystal violet stained colonies 8 days after treatment (**A**). Reliable counting of the colonies was based on dividing the plate areas into segments (**B**). Statistical analysis revealed a significant reduction after R+mEHT treatment compared to the control group (*p* = 0.0195) (**C**). Kruskal–Wallis test also showed a significant difference among the 4 groups (*p* = 0.0064) TER = 3.1. The asterisk (*) marks the value of *p* ≤ 0.05.

**Figure 4 ijms-21-05100-f004:**
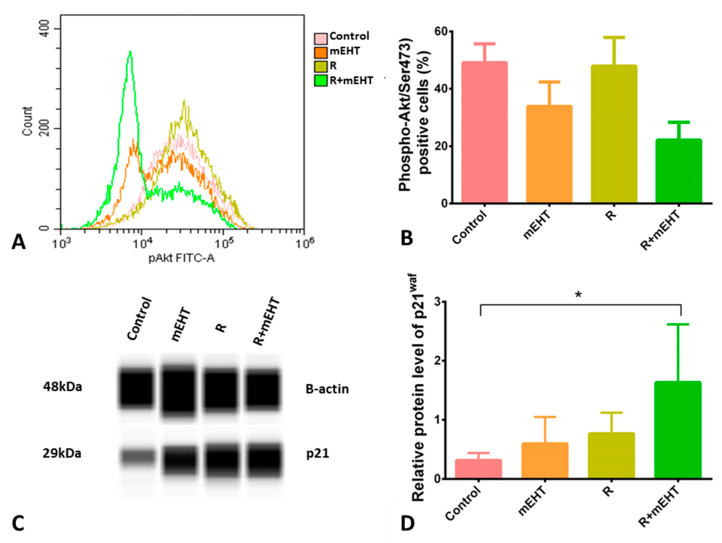
Treatment induced reduced p-Akt/Ser473 and elevated p21^waf1^ levels in Panc1 cultures 24 h post-treatment. Flow cytometry histogram (**A**) and quantitation graph of p-Akt/Ser473 levels. Kruskal–Wallis test revealed a significant difference (*p* = 0.015) among the 4 groups (**B**). p21^waf1^ protein expression detected in Western blots (**C**). Kruskal–Wallis test (*p* = 0.049) revealed a significant difference among groups. Dunn’s post-hoc test confirmed a significant increase in p21^waf1^ level in the R+mEHT group compared to the control (*p* = 0.045) (**D**). The asterisk (*) marks the value of *p* ≤ 0.05.

**Figure 5 ijms-21-05100-f005:**
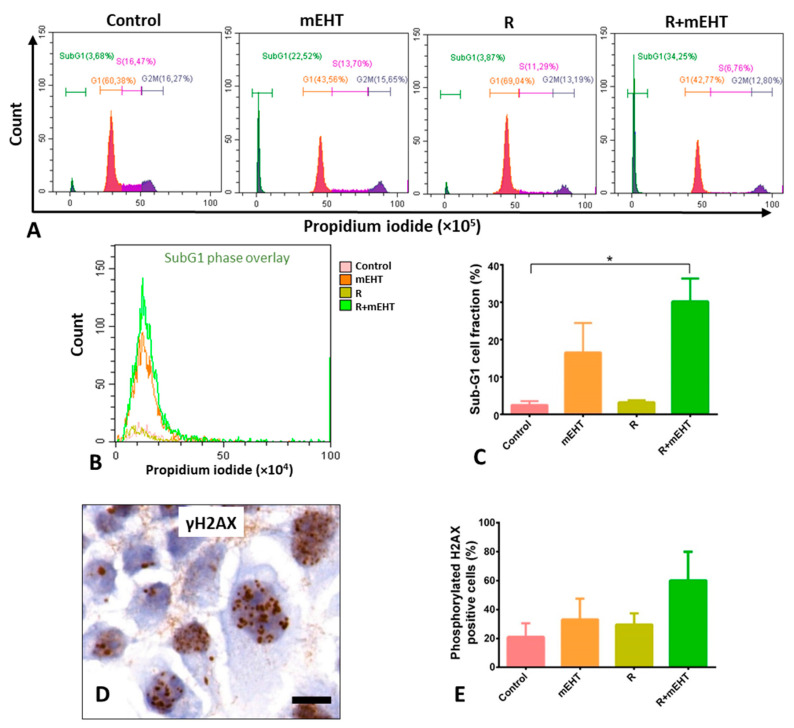
Treatment related elevated subG1-fractions and moderate increase in nuclear phosphorylated H2AX protein levels in Panc1 cultures 24 h post-treatment. Representative flow cytometry graphs of cell cycle fractions (**A**). A summary (**B**) and graphical representation of subG1-phase fractions (**C**) showing strong significance among the 4 groups (*p* = 0.0006), which was confirmed with the Dunn’s test between R+ mEHT treatment and the controls (*p* = 0.028). Upregulated granular γH2AX foci in tumor cell nuclei after combined R+mEHT treatment (**D**, scale bar: 10 µm) and their graphical representation after quantification (**E**), whose difference, however, did not reach significance compared to the controls. The asterisk (*) marks the value of *p* ≤ 0.05.

**Figure 6 ijms-21-05100-f006:**
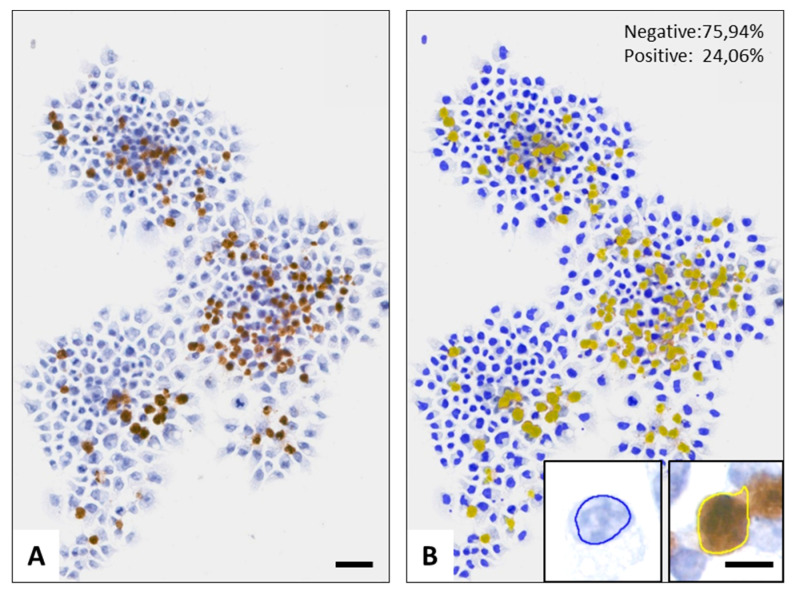
Demonstration of digital image analysis using the cleaved caspase-3 immunoreaction in R+mEHT treated Panc1 culture as an example. The NuclearQuant module of the QuantCenter software allowed the segmentation and counting of brown immunopositive cells (**A**, scale bar: 50 µm) and to determine their proportion to the whole population stained using hematoxylin. The accurately selected masque highlights positive (yellow labels) and negative (blue labels) cells shown also at high power in insets (**B**, scale bar: 10 µm).

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
