# Peer review of "Modulated Electro-Hyperthermia Resolves Radioresistance of Panc1 Pancreas Adenocarcinoma and Promotes DNA Damage and Apoptosis In Vitro"

_ijms, 2020, doi:10.3390/ijms21145100_

Round 1

Reviewer 1 Report

The manuscript contains new experimental results related to the combined effect of modulated electro-hyperthermia and radiotherapy on Panc1 pancreas ductal adenocarcinoma cells. These results are very important, because they can be useful for development of effective scenarios of pancreatic ductal adenocarcinoma therapy. They can also be useful for other fields of medicine and biology. The manuscript can be published after minor revision the necessity of which results from the following:

  1. The abbreviation BSD (page 2, line 79) should be explained. A minor erratum on the same line should be corrected (“13,56” should be replaced by “56”).
  2. The sentence on page 2, lines 83–85, should be edited, because the use of the word “accumulated” for description of electric field is unusual.
  3. The potentially important phrase “the electric field and the concomitant heat can synergise” (page 8, lines 204–205) is unclear, because the authors do not mention established and/or supposed nonthermal electric field effect(s) which could be important in the situations under consideration. Such effects were discussed in literature (see, e.g., J.D.T. Arruda-Neto et al., Int. J. Rad. Biol. 85, 314 (2009); M.L. Shmatov, Phys. Part. Nucl. Lett. 14, 533 (2017)), but the possibility of their manifestation in different situations, in particular, in the experiments described in the manuscript, require special studies.

Author Response

The instrument’s name BSD (page 2, line 79) is replaced by more precise definition of hyperthermia delivery: “These data led to the initiation of clinical trials combining PDAC chemotherapy with locoregional hyperthermia generated by instruments using different radiofrequency and power ranges based either on radiative (70-120 MHz, 800-1500W, 60 min/treatment) [39] or capacitive (mEHT: 13.56 MHz, 60-150 W, 60 min/treatment) [40] delivery.” 

The Hungarian version of “13,56” value is corrected to the English version13.56.

The word “accumulated” used for the electric field is replaced to “concentrated” in malignant tissues.

We explained the criticized sentence “the electric field and the concomitant heat can synergise” (page 8, lines 204–205). Since the references suggested by the referees use static electric field, for an explanation of the additional non-thermal effects of mEHT using alternating electric field, we inserted a sentence into our revised version referring to 2 studies by Andocs and co-workers, which suggest that “mEHT can selectively deposit energy onto the cell membrane” which may deliver additional irreversible stress to tumor cells (Andocs et al. Cell Death Discov. 2016 Jun 13;2:16039; Yang et al. Oncotarget 2016 Dec 20;7(51):84082-84092.)

Reviewer 2 Report

It is an interesting paper , very complex and well done, you have to reduce the introduction and reduce it because a little ripetitive. Take off the oldest references and reduce the total number of them. Try to have the most related citations to pancreatic tumors and the increase of results that you can obtain adding hyperthermia to standard care. Specify better TER in your experiments  I mean about therapeutic enhacement ratio 

Author Response

Some repetitive information and older references were eliminated from the reviesed version of the manuscript as recommended. Furthermore, some additional references were included in relation to the topic how hyperthermia can contribute to the standard oncotherapy of pancreatic tumors. TER (therapy enhancement ratio) was precisely defined and calculated compared to the results of irradiation at body temperature.